# Experience of Ethnic Discrimination, Anxiety, Perceived Risk of COVID-19, and Social Support among Polish and International Students during the Pandemic

**DOI:** 10.3390/ijerph20075236

**Published:** 2023-03-23

**Authors:** Anna Bokszczanin, Olga Gladysh, Anna Bronowicka, Marek Palace

**Affiliations:** 1Institute of Psychology, University of Opole, 45-052 Opole, Poland; 2Institute of Psychology, Polish Academy of Sciences, 03-378 Warsaw, Poland; 3School of Psychology, Liverpool John Moores University, Liverpool L3-5UG, UK

**Keywords:** ethnic discrimination, anxiety, international students, COVID-19 pandemic, social support, university students, Poland

## Abstract

Background: Our research aimed to assess the experiences of ethnic discrimination among students in Poland (Polish and international) during the COVID-19 pandemic. We also tested the prevalence of anxiety symptoms and their relationship with perceived COVID-19 risk, the severity of discrimination, and social support. Methods: The data from Polish (*n* = 481) and international university students (*n* = 105) were collected online (November–January 2020). Participants completed measures of ethnic discrimination (GEDS), anxiety scale (GAD-7), COVID-19 risk perception index, and perceived social support scale (MSPSS) questionnaires. Results: The results showed that international students reported being much more discriminated than Polish students during the first year of the COVID-19 pandemic. Contrary to our expectation, a higher risk of anxiety disorders (GAD) was observed in 42% of Polish students compared to 31% of international students. The predictors of higher anxiety symptoms among both groups were the perceived risk of COVID-19 and the greater severity of ethnic discrimination. In both groups, the perceived social support had a protective role in anxiety symptomatology. Conclusions: The high prevalence of discrimination, especially among international students, simultaneously with high symptoms of anxiety, requires vigorous action involving preventive measures and psychological support.

## 1. Introduction

Data collected in various countries worldwide document the negative impact of the COVID-19 pandemic on the mental health of university students [1,2,3,4,5]. Rapid changes in learning and lifestyle are accompanied by severe stress. Particular stressors contributing to mental health deterioration are remote learning, maintaining physical distance, job loss, a sedentary lifestyle, skipping meals, not taking care of order and hygiene, and many others [6,7,8]. High indicators of psychopathological symptoms fundamentally resulted from the reaction to numerous stressors the students experienced [4,5,9].

Many studies have focused on assessing depression and anxiety symptoms [10,11,12,13]. A meta-analysis of cross-sectional studies on anxiety and depression in China showed that the overall prevalence of anxiety and depression symptoms in students during the COVID-19 pandemic was 24% and 22%, respectively [12]. Researchers also noted increases in anxiety and depression in European countries. For example, 46.0% of students in Portugal were at risk for anxiety disorders in 2019, before the pandemic, and 64.5% one year later in 2020 [11]. Significant changes in the intensity of harmful mental health symptoms, anxiety, and depression were observed at the beginning of the pandemic in Switzerland [10]. Student participants showed a significant decrease in moderate-to-severe anxiety scores of 20.2% and 15.6% at T0 and T1, respectively. Polish students reported a high risk of anxiety disorder, which oscillated from 31% to 46% depending on the coronavirus pandemic wave [13]. The severity of anxiety symptoms varied depending on geographical location, measurement instrument, and time of measurement [14].

A dominant factor directly related to the pandemic and predicted high anxiety levels was the threat of and uncertainty about the effects of the rapidly spreading viral disease [1,6,12,15]. Similar results were found in numerous studies, where factors related to the perceived risk of the COVID-19 epidemic were significantly associated with the risk of psychiatric disorders [16,17,18].

In turn, an essential factor protecting mental health is perceived social support. Through contact with other people and the support of friends, family, and acquaintances, the skills for effectively coping with stress and perceiving a stressful situation as less severe increase [19,20]. For college students, the loss of social interaction and campus life during the pandemic was severe [21]. However, students resistant to stress had a developed social support network and a strong sense of support from others and were protected against depression, loneliness, and anxiety [19,20,21].

International students are particularly vulnerable to the pandemic’s adverse effects, which is documented by several studies [22]. Poland had about 82,000 international students at the beginning of the pandemic, representing 6.8% of all students; half of these were of Ukrainian origin [23]. However, international students studying in Poland have yet to receive much attention regarding mental health predictors and their impact during the pandemic [24,25].

The primary source of distress for international students is feeling rejected by people in their host country. For example, Chavajay and Skowronek [26] reported that 48% of a sample of 130 international students from 33 countries studying in the US experienced discrimination by a member of the host culture. International students often encounter discrimination because of a foreign-sounding accent or a different appearance in places such as a supermarket or university [27]. For example, one study in Poland found that 61.2% of medical students from Asia have experienced prejudice related to the current coronavirus epidemic [28]. Discrimination’s practical manifestations concern various behaviors, such as stepping away, staring continuously, making xenophobic comments, and opening the doors with a tissue after an Asian student has touched the handle. The surveyed students reported that these experiences hurt their well-being. Besides the general public and Polish students, the perpetrators of discrimination were also academic lecturers. Discriminatory behaviors toward international students were also encountered in Poland before the pandemic. Bilewicz’s team [24,25] observed that a third of international students said they had received insults or offensive comments. These students described a worsened mood and often showed PTSD symptoms. The study’s results displayed that more frequent exposure to hate speech, one of the dimensions of discrimination, was associated with lower levels of positive affect, lower life satisfaction, higher levels of PTSD, and greater negative affectivity among students. The existing data confirmed that experiences of discrimination are associated with severe psychological and social negative consequences for international students [29,30].

Although only 1% of Poland’s residents report belonging to an ethnic minority, in public opinion polls, the respondents distinguish over 20 different minorities, mainly guided by the region of residence or the characteristics of the dialect [31]. Social research and reports from nongovernmental organizations indicate a significant scale of discrimination, prejudice, and stereotypical ideas about various social groups in Poland [32]. A study conducted among two minority groups, Kashubians and Silesians, during the COVID-19 pandemic, showed that these groups suffer from ethnic discrimination and marginalization. Silesians, an ethnic minority not recognized by the Polish state, were additionally stigmatized during the pandemic because the largest outbreak of COVID-19 occurred in this region, with the largest concentration of Polish residents [33]. Therefore, we expected that Polish university students may also experience ethnic discrimination due to their region, place of residence, or specific accent. Hence, we adopt a broad definition of ethnic discrimination in our study.

The experience of ethnic discrimination is conceptualized as a type of stress leading to significant psychological and even physiological reactions (e.g., hypertension). It increases the likelihood of passive and other maladaptive forms of coping with stress (e.g., anger) [29,30]. Many studies in Poland have focused on the student population, but few have focused on discrimination issues. The studies published at that time focused on high rates of anxiety, the explanation of which we sought primarily in the situation of a pandemic. Perceived social support was chosen because it is a well-known factor protecting against stress.

Therefore, our study aimed to examine the prevalence of ethnic discrimination among students in Poland (international and Polish) during the COVID-19 pandemic. We also examined the prevalence and intensity of anxiety symptoms and their relationship with risk factors, including perceived COVID-19 risk, the severity of ethnic discrimination, and the protective role of perceived social support.

## 2. Materials and Methods

### 2.1. Study Design and Procedure

The study was based on data from a cross-sectional online survey prepared using Google Forms. The survey in Poland (November 2020 to January 2021) was directed at all university students, including international, from every research discipline. The online questionnaire was delivered in Polish and English so that international students who do not speak Polish well could complete it. However, only eight (1.4%) people completed the survey in English. Our measurement tools had proven psychometric properties in Polish and English [15,34,35,36]. To reach as many Polish and international students as possible, the researchers and several engaged students from the science club used various recruitment strategies, such as sharing a link to the survey on social media, mailing lists for student groups, and sending links to employees of other universities with a request to disseminate. Participants had to give informed consent before participating; at the end, they were asked to send the survey link to their friends.

### 2.2. Statistical Analysis

First, we examined means (*M*), standard deviations (*SD*), and percentage proportions in terms of the sociodemographic characteristics of the study participants.

Next, we estimated the percentage prevalence of ethnic discrimination in the two groups according to the established measuring tool [29].

We used chi-square values and Sommer’s d-effect sizes to compare Polish and international student groups separately for each item of the General Ethnic Discrimination Scale (GEDS).

The severity of anxiety symptoms was assessed using the GAD-7 scale [35]. Next, we used the one-way ANOVA ***F*** tests to compare the Polish and international student groups’ variances regarding continuous variables: the severity of ethnic discrimination, anxiety symptoms, perceived risk of COVID-19, and perceived social support. The measure of effect size is (partial) eta squared (η^2^).

We also compared the response scores to each *COVID-19 risk perception question* and differences in the groups, measured by the Mann–Whitney U test with eta (total) effect size (η).

The hierarchical regression analysis was performed separately among each group of the study participants; the aim was to identify the significant predictors of anxiety symptom severity characteristics in Polish vs. international students. 

All statistical analyses were performed with IBM Statistics, version 28.

### 2.3. Measures

#### 2.3.1. Ethnic Discrimination

The General Ethnic Discrimination Scale (GEDS) is an 18-item self-report inventory of recognized ethnic discrimination used in health research with any ethnic group. The GEDS results were related to psychiatric symptoms among people of ethnic minorities and white people, with the stronger relationships for people of ethnic minorities [29]. The scale reported good internal consistency (94–95), and additionally, the GEDS has a confirmed equal effectiveness in modeling the factor structure of perceived ethnic discrimination across diverse racial and ethnic groups [29]. The scale is composed of three unidimensional subscales: (1) Recent Discrimination; (2) Lifetime Discrimination; (3) Appraised Discrimination. However, this study only used the Recent Discrimination scale, relating to the participants’ experience in the last years. For this purpose, 18 items were translated into Polish and back-translated as required [34]. Respondents rate the frequency of perceived discriminatory and racist events during the past year on a 6-point Likert-type scale: 1 = Never; 2 = Once; 3 = Sometimes; 4 = A Lot; 5 = Most of the Time; and 6 = Almost All the Time. The range for the whole scale was 18 to 108. Higher scores indicate a higher level of ethnic discrimination. The Cronbach’s α for the GEDS in this study was 0.93.

#### 2.3.2. Anxiety

The severity of anxiety symptoms was assessed using the 7-item generalized anxiety disorder (GAD-7) scale [35]. The GAD-7, a brief measure of anxiety symptoms, is a popular scale to measure the generalized level of persistent worry according to DSM-5 criteria. Respondents rate the frequency of their experienced anxiety during the last two weeks on a 4-point Likert-type scale: 0 = Not at all; 1 = Several days; 2 = More than half the days; and 3 = Nearly every day. A higher score indicates a higher general anxiety disorder (GAD) risk. The range for the whole scale was 0 to 21. The Cronbach’s α for the GAD-7 in this study was 0.92.

#### 2.3.3. COVID-19 Risk Perception

The COVID-19 risk perception index is a general risk-perception measure, covering affective, cognitive, and temporal–spatial dimensions and including six items relating to respondents’ perceived severity of the pandemic over the next six months [26]. The items concern worry about the virus, the perceived likelihood of contracting the COVID-19 virus, and the likelihood of their family and friends catching it. Respondents rate the frequency of the perceived risk of COVID-19 on a 7-point Likert scale (three items: 1, 2, 3) or a 5-point scale (three items: 4, 5, 6). Item examples: 1. How worried are you personally about the following issues at present?-Coronavirus/COVID-19? From 1 = Not at all worried to 7 = Very worried. 5. How much do you agree or disagree with the following statements?-I will probably get sick with the coronavirus/COVID-19. From 1 = strongly disagree to 5 = strongly agree. The total risk perception measure is calculated as the mean value of all six items, ranging from 0 to 36. A higher score indicates a higher perceived risk of COVID-19. The Cronbach’s α for this measure in this study was 0.60. 

#### 2.3.4. Perceived Social Support

Perceived Social Support was measured using the Multidimensional Scale of Perceived Social Support (MSPSS) [36]. It is a 12-item scale comprising perceived social support from family, friends, and significant others. Each item is measured on a Likert-type scale ranging from 1 to 7: 1 = Very strongly disagree to 7 = Very strongly agree. Each subscale item relates to practical help, emotional support, availability to discuss problems, and help in decision making. The range for the whole scale was 12 to 84, showing that the higher the score, the higher the perceived social support. The Cronbach’s α for this measure in this study was 0.93.

#### 2.3.5. Sociodemographic Variables

The following sociodemographic characteristics are included in the analyses: gender (male/female), age in years, ethnic origin, subject of study, year of study, work, and student status (Polish vs. international).

## 3. Results

### 3.1. Participants

The data include a total of 586 university students, 481 Polish and 105 international, currently studying in Poland. The group of foreign students included 67 Ukrainian (64.8%), 8 Vietnamese (7.6%), 4 German (3.8%), and 2 Belarusian (1.9%) students. In addition, our data included individuals from Ethiopia, Nigeria, Greece, Mexico, Iran, Palestine, Czechia, and Russia (7.6%). Due to missing data, we were unable to determine the origin of 15 (14.3%) foreign students. Table 1 presents the sociodemographic characteristics of the study participants. Participants’ ages were primarily in the range of 17–22 (81%), and most (70%) were women. Most Polish participants studied humanities and social sciences (26% and 19%, respectively). Most international students studied economics and technical studies (28% and 27%, respectively). The majority of our respondents were undergraduates (Polish 90%; international 67%) and were not currently employed (67% vs. 57%).

### 3.2. Discrimination Experiences

The authors of the GEDS scale suggest that the discrimination scores can be re-coded into the lowest, middle, and highest thirds. These were scores ≤ 20, 21–27, and >28 [29] (p. 87). Figure 1 demonstrates the proportion of individuals who experienced ethnic discrimination in the first year of the COVID-19 pandemic, broken down into three levels. The cross-Table 2 × 3 (two groups of students vs. three intervals) showed that the difference between Polish and international students was statistically significant, chi-square = 22.76 (*df* = 2), *p* < 0.001, and small Sommer’s d effect size = 0.14. Low levels of discrimination were reported by 62.1% (*n* = 364) of Polish students and 43.8% (*n* = 46) of international students. A higher percentage of discrimination at the middle level was also observed among international students, 26.7% (28), compared to Polish students, 20.8% (*n* = 100). A much higher percentage of international students, 29.5% (*n* = 31), perceived high levels of discrimination compared to Polish students, 13.1% (*n* = 63).

Table 2 presents the means, standard deviations, chi-square values, and Sommer’s d-effect sizes. Differences between the Polish and international students were calculated separately for each of the 18 items of the Discrimination Scale (GEDS) listed in the table. We used the cross-table to calculate each item’s chi-square value (*df* = 5) and Sommer’s d-effect size. We can see that the severity of discrimination against international students occurred significantly more frequently, i.e., in 14 out of 17 listed items, than against Polish students. A positive and small effect size characterized the observed differences. No statistically significant differences were found in terms of the three listed items (6, 9, 16) between the two analyzed groups.

### 3.3. Prevalence of Anxiety

In further calculations, we focused on estimating the risk of a generalized anxiety disorder (GAD) as measured by the GAD-7 scale [35]. It turned out that in the high-risk group (GAD-7 ≥ 10), there is a much higher percentage of Polish (42%, *n* = 202) than foreign students (31%, *n* = 33). This difference is statistically significant *p* < 0.02 (*Chi-square* = 4.01, *Phi* = 0.08).

### 3.4. Comparison of Polish and International Students for Study Variables

Table 3 compares Polish and international students using the one-way ANOVA test regarding the severity of anxiety symptoms, discrimination, perceived risk of COVID-19, and perceived social support. It shows that international students felt less anxiety than Polish students but were much more discriminated against during the last year. Furthermore, international students perceived a statistically significantly lower risk of coronavirus than Polish students. A small effect size characterized all differences observed. There were no differences in perceived social support between the analyzed groups of respondents.

Since the study was carried out during the COVID-19 pandemic, it was interesting to examine the detailed differences regarding the sense of threat related to the pandemic between groups of students. To this end, we compared the average response scores to each question of the COVID-19 risk index. Table 4 shows the means, standard deviations, and differences measured by the Mann–Whitney U test for each of the six questions. The presented results show significant differences with a medium effect size between the assessment of Polish and international students regarding possible coronavirus cases in the country where they are currently staying. International students rated the risk of seriously getting sick with the coronavirus significantly higher than Polish students.

### 3.5. Hierarchical Regression Analyses

Separate regression analyses were conducted due to the expected differences between Polish and international student groups in terms of explanatory variables and the percentage of explained total variance. Therefore, two hierarchical regression equations were performed separately for each group to examine the effects of gender, age, perceived risk of COVID-19, perceived ethnic discrimination, and perceived social support for anxiety symptoms (Table 5). Gender and age were introduced in the first step to check what percentage of the variance of the dependent variable is related to the constant characteristics of the respondents. We then introduced two independent variables into the equation, the perceived risk of COVID-19 and perceived ethnic discrimination, which were defined as factors related to the risk of anxiety (stressors) depending on the external situation. In the third and final step, we introduced a variable measuring perceived social support to check what percentage, above and beyond the other variables, is associated with social support, which we define as a protective factor.

All predictors entered into the equations explained the statistically significant part of the variance in anxiety symptoms both in the group of Polish (*R*^2^ = 0.22, *p* < 0.001) and international students (*R*^2^ = 0.29, *p* < 0.001). All independent variables separately predicted a significant part of the variance of the dependent variable in the Polish group. Female gender, younger age, higher perceived COVID-19 risk, and higher severity of ethnic discrimination were associated with greater anxiety symptoms. In the group of international students, greater severity of anxiety was associated only with a greater perceived risk of COVID-19 risk and greater severity of discrimination. In both groups, the lower the perceived social support, the greater the severity of anxiety observed in students.

## 4. Discussion

Our study aimed to examine the prevalence of ethnic discrimination among Polish and international students during the COVID-19 pandemic. The results of the online survey across universities in Poland confirmed that a high percentage of students report experiencing ethnic discrimination during the first year of the pandemic. Moreover, we observed that the severity of discrimination was much higher towards foreign students than towards Polish ones. Similar results were collected in other countries. Attention is drawn to the sense of cultural disconnection and to facing the prevailing sociopolitical climate [6,7,37,38,39]. However, the main grounds of discrimination are considered to be foreign status, speaking with a foreign accent, and belonging to a visible racial or ethnic minority. No matter the reason, experiencing discrimination harms mental health, lowers self-esteem, and hinders international students’ acculturation [22,25,28,40,41,42]. The ongoing COVID-19 pandemic is believed to have increased stigma and prejudice against people of foreign nationalities due to the lack of knowledge about the COVID-19 virus and the fear of the consequences. Fear about the virus has caused the social need to blame someone for the pandemic. The psychological process of scapegoating is a control mechanism updated in the face of medical uncertainty [42,43,44].

The objective of our study was also to assess the prevalence of anxiety symptoms and its predictors among Polish and international students. Our study shows that anxiety levels among the surveyed students were high, similar to the results of other studies during the pandemic [4,5,8,9,10,11,12,13,14,16,18]. However, contrary to our expectations, the risk of a generalized anxiety disorder (GAD) was significantly higher in Polish than in international students (42% vs. 31%, respectively). This result seems paradoxical; however, the data suggest that international students can cope with stress better than Polish students. Integrating with a group of similar people (i.e., other international students) may protect their mental health and reduce the exacerbation of harmful anxiety symptoms. Research indicates that solid intragroup ties could reduce anxiety during the pandemic and serve as a “social medicine” [24].

International students perceived the threat of the spreading viral disease as greater than Polish students did. It can be assumed that among international students, the fear of COVID-19 and its consequences was related to a greater misunderstanding of the pandemic situation and the counteractive actions by various responsible institutions in Poland. In addition, the policy of counteracting the pandemic in different countries worldwide was significantly distinct, which could also increase uncertainty and a sense of threat [44,45]. Greater fear of the coronavirus may have been also intensified by the separation of international students and distance from family and loved ones [1,2,3,5]. It can be supposed that the sense of danger was related to the excessive death rate in the first year of the pandemic. The high death rate in 2020 in Poland, compared to the 2016–2019 average, was 194 per 100,000 inhabitants. Therefore, it was a tragic year, especially in the region of Silesia, a precise local “hot spot” of the pandemic [33,46], where the University of Opole is located.

As expected, the significant predictors of the severity of anxiety symptoms among Polish and international students were stressors related to the perceived risk of COVID-19 and discrimination experiences. Thus, our study confirmed the role of stress factors in the development of anxiety in students caused by the pandemic [10,13,14,15]. The study results also emphasized the harmful mental health role of the stress of discrimination among students, which has not been often reported by the Polish researchers, even before the pandemic [47,48].

Finally, our outcomes suggest that a generalized sense of social support during the pandemic alleviated symptoms of anxiety among university students. Researchers from other countries observed comparable results [19,20]. The findings confirm the existing knowledge that the perception of support from family and friends improves coping effectiveness, reduces distress, improves mood, and buffers the development of psychopathology symptoms [21]. Most of our respondents continued their education in family homes during the pandemic, which made direct contact with friends and acquaintances difficult. These difficulties were partly overcome through contacts in the network [47,48,49,50].

### Limitations

One limitation of this study is its cross-sectional nature, making it impossible to infer causality. However, the biggest drawback is the small number of international students, which makes it challenging to draw generalized conclusions. In the subsequent study, other approaches should be used to encourage international students to participate in the survey. Additionally, more women than men participated in the survey, making it difficult to draw conclusions about the entire student population. The online survey may have also attracted people with psychological problems during the pandemic who wanted to share them with researchers. The disadvantage of the online survey methodology is the need for more control over who completes it. Moreover, only self-report questionnaires were used. Therefore, our results only indicate risk and cannot replace a reliable psychological diagnosis. The limitations of the quantitative study did not allow us to collect in-depth, detailed data on student discrimination. In the future, qualitative research is proposed to more accurately answer our research questions, so we can gain better insights into the issues explored.

## 5. Conclusions

In summary, the results of the presented study confirmed some existing findings regarding the mental health of university students in Poland during the coronavirus pandemic revealed new relationships and neglected research areas, and led to important conclusions. First, far more support is needed from universities hosting foreign students [47]. The high prevalence of ethnic discrimination experiences, especially among international students, and the high level of anxiety symptoms require vigorous action. Among the most important trends nowadays is the so-called student exchange between countries and internationalization of universities. From this perspective, it seems obvious that students moving to a foreign country need help adapting to the new culture [48,49]. Positive adaptation to a foreign culture is conducive to students’ well-being. Thus, introducing more antidiscriminatory educational programs for universities, including students and lecturers, is recommended. Well-designed programs improve the understanding of the problems faced by students, help them cope with discrimination and prejudice better, and allow them to enjoy all aspects of student life [50].

## Figures and Tables

**Figure 1 ijerph-20-05236-f001:**
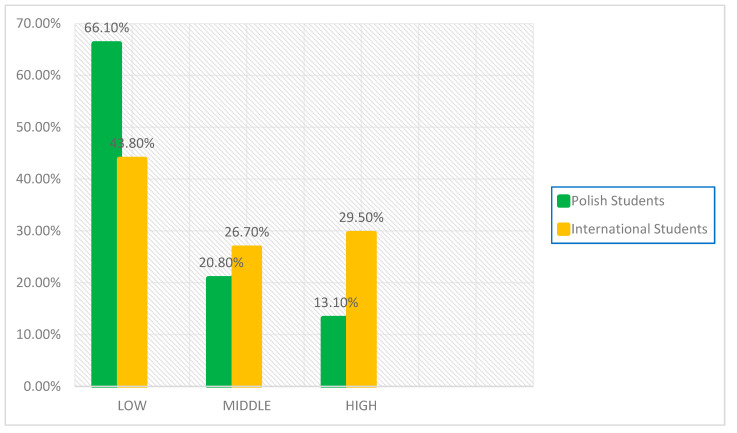
The proportion of perceived ethnic discrimination in the group of Polish (*n* = 481) and international university students (*n* = 105) by the General Ethnic Discrimination Scale (GEDS); Responses in the range LOW = 1 to 20, MIDDLE= 21 to 27, HIGH => 28; chi-square = 22.76 (*df* = 2), *p* < 0.001, Sommer’s d effect size = 0.14.

**Table 1 ijerph-20-05236-t001:** Sociodemographic characteristics of the study participants.

	All (*n* = 586)		Polish Students (*n* = 481)		International Students (*n* = 105)	
	*n*	%	*n*	%	*n*	%
Gender						
Female	374	70.0	346	72.2	63	60.0
Male	160	30.0	133	27.8	42	40.0
Age						
17–19	125	23.4	111	23.2	22	21.0
20–22	271	50.7	242	50.5	57	54.3
23–25	101	18.9	93	19.4	17	16.2
26 and more	37	6.9	33	6.9	9	8.5
Subject of study						
Social Sciences	100	17.2	92	19.1	8	7.6
Humanities	148	24.7	125	26.0	23	21.9
Economics	103	17.6	68	14.1	35	28.3
Artistic Sciences	22	3.8	21	4.4	1	1.0
Medical Sciences	76	12.8	68	14.1	8	7.6
Technical Studies	60	10.2	32	6.7	28	26.7
Other	77	13.7	75	15.6	2	1.9
Year of study						
1	237	40.4	219	45.5	18	17.1
2	126	21.5	101	21.0	25	23.8
3	93	15.9	65	13.5	28	26.7
4	76	13.0	59	12.3	17	16.2
5	54	9.2	37	7.7	17	16.2
Work						
Does not work	380	64.8	320	66.5	60	57.1
Part-time work	79	13.5	58	12.1	21	20.0
Full-time work	62	10.6	49	10.2	13	12.4
Looking for a job	65	11.1	54	11.2	11	10.5

**Table 2 ijerph-20-05236-t002:** Differences in discrimination behaviors perceived by the Polish and international students.

	Polish Students (n = 481)		International Students (n = 105)		Chi-Square-Value	Sommer’s d-Value
	M	SD	M	SD		
1. How often have you been treated unfairly by teachers and professors because of your home origin/accent/ethnicity?	1.25	0.68	1.48	0.75	17.49 **	0.13 **
2. How often have you been treated unfairly by your employers, bosses and supervisors because of your home origin/accent/ethnicity?	1.27	0.70	1.58	0.77	17.43 **	0.15 **
3. How often have you been treated unfairly by your co-workers, fellow students and colleagues because of your home origin/accent/ethnicity?	1.37	0.84	1.59	0.88	10.83 **	0.10 **
4. How often have you been treated unfairly by people in service jobs (by store clerks, waiters, bartenders, bank tellers and others) because of your home origin/accent/ethnicity?	1.26	0.62	1.58	0.71	22.29 ***	0.13 **
5. How often have you been treated unfairly by strangers because of your home origin/accent/ethnicity?	1.26	0.66	1.63	0.73	35.63 ***	0.18 ***
6. How often have you been treated unfairly by people in helping jobs (by doctors, nurses, psychiatrists, case workers, dentists, school counsellors, therapists, social workers and others) because of your home origin/accent/ethnicity?	1.28	0.74	1.30	0.74	2.21	0.02
7. How often have you been treated unfairly by neighbours because of your home origin/accent/ethnicity?	1.16	0.62	1.32	0.67	11.94 *	0.15 *
8. How often have you been treated unfairly by institutions (schools, universities, law firms, the police, the courts, the Department of Social Services, the Unemployment Office and others) because of your home origin/accent/ethnicity?	1.28	0.70	1.60	0.81	15.72 **	0.10 *
9. How often have you been treated unfairly by people that you thought were your friends because of your home origin/accent/ethnicity?	1.29	0.75	1.28	0.76	4.01	−0.02
10. How often have you been accused or suspected of doing something wrong (such as stealing, cheating, not doing your share of the work or breaking the law) because of your home origin/accent/ethnicity?	1.14	0.52	1.31	0.60	9.84	0.12 *
11. How often have people misunderstood your intentions and motives because of your home origin/accent/ethnicity?	1.23	0.64	1.42	0.67	15.25 **	0.13 **
12. How often did you want tell someone that they treat you unfairly because of your home origin/accent/ethnicity, but chose to say nothing?	1.15	0.55	1.53	0.65	41.46 ***	0.29 ***
13. How often have you been really angered by some discrimination based on your home origin/accent/ethnicity?	1.17	0.58	1.53	0.68	36.87 ***	0.25 ***
14. How often have you been forced to take drastic steps (such as filing a grievance, filing a lawsuit, quitting your job, moving away and other actions) so as to cope with discrimination based on your home origin/accent/ethnicity?	1.10	0.50	1.23	0.53	18.65 **	0.17 **
15. How often have you been verbally abused because of your home origin/accent/ethnicity?	1.09	0.42	1.25	0.48	16.12 **	0.17 **
16. How often have you gotten into an argument or a fight about something racist that was done to you or another member of your home origin/accent/ethnicity?	1.15	0.56	1.15	0.54	4.14	0.05
17. How often have you been made fun of. picked on, pushed, shoved, hit, or threatened with harm because of your home origin/accent/ethnicity?	1.12	0.50	1.27	0.55	16.44 **	0.17 *
18. How different would your life be now if you HAD NOT BEEN treated in a racist and unfair way?	1.37	1.01	1.78	1.07	45.30 ***	0.20 ***

Note. *** *p* < 0.001, ***p* < 0.01, **p* < 0.05.

**Table 3 ijerph-20-05236-t003:** Means, standard deviations, and one-way ANOVA for the Polish and international student groups for study variables.

Variables	Polish Students (n = 481)		International Students (n = 105)		F	(Partial) η^2^
	M	SD	M	SD		
GAD-7	9.07	5.878	7.52	5.20	6.23 **	0.01
GEDS	21.93	7.92	25.83	10.54	18.40 ***	0.03
COVID-19 Risk	3.77	1.09	4.30	1.36	18.48 ***	0.03
MSPSS	59.93	16.44	61.76	16.70	1.07	0.00

Note. ** *p* < 0.01, *** *p* < 0.001. GAD-7 = General Anxiety Disorder; COVID-19 Scale = COVID-19 Risk Perception Scale; MSPSS = Multidimensional Scale of Perceived Social Support; GEDS = General Ethnic Discrimination Scale.

**Table 4 ijerph-20-05236-t004:** Means, standard deviations and Mann–Whitney U test for the Polish and international student groups for the COVID-19 risk index.

Variables	Polish Students (*n* = 481)		International Students (*n* = 105)		U *p*-Value	η^2^
	M	SD	M	SD		
1. How worried are you personally about the following issues at present? - Coronavirus/COVID-19	3.76	1.698	4.00	1.840	23,200.000	0.01
2. How likely do you think it is that you will be directly and personally affected by the following in the next 6 months? - Catching the coronavirus/COVID-19?	3.71	2.009	3.75	2.023	24,955.500	0.00
3. How likely do you think it is that your friends and family in the country you are currently living in will be directly affected by the following in the next 6 months? - Catching the coronavirus/COVID-19?	4.55	1.852	4.56	1.860	25,232.500	0.01
4. The coronavirus/COVID-19 will NOT affect very many people in the country I’m currently living in? ^R^	2.15	1.170	3.80	1.274	9,308.000 ***	0.22
5. I will probably get sick with the coronavirus/COVID-19?	3.19	1.335	2.93	1.368	22,459.500	0.01
6. Getting sick with the coronavirus/COVID-19 can be serious?	3.71	1.228	3.97	1.164	22,167.500 *	0.01

Note. ^R^ = Reverse-coded; *** *p* < 0.001, * *p* < 0.05.

**Table 5 ijerph-20-05236-t005:** Hierarchical regression analysis of variables predicting anxiety symptoms (GAD-7).

Predictors		Polish Students (*n* = 481)				International Students (*n* = 105)			
		b	SE b	β	t	b	SE b	β	t
Step 1	Constant	8.33	1.284	0.09	6.49 ***	−0.27	3.94		−0.07
	Gender ^1^	1.22	0.32	−0.09	2.00 *	1.47	1.02	0.13	1.44
	Age	−0.64	1.284		−2.00 *	0.25	0.16	0.15	1.55
Step 2	Constant	1.69	1.615		1.05	−3.75	4.01		−0.94
	Gender ^1^	1.11	0.58	0.09	1.92	1.13	0.97	0.11	1.16
	Age	−0.72	0.31	−0.10	−2.33 *	0.13	0.16	0.08	0.85
	COVID-19 risk	0.85	0.24	0.16	3.56 **	0.69	0.36	0.18	1.95 *
	GEDS	0.17	0.03	0.23	5.32 ***	0.13	0.05	0.28	2.92 **
Step 3	Constant	9.45	1.74		5.43 ***	8.63	4.65		1.85
	Gender ^1^	1.69	0.54	0.13	3.12 **	1.63	0.90	0.15	1.79
	Age	−0.62	0.29	−0.09	−2.15 *	−0.05	0.15	−0.03	−0.32
	COVID-19 risk	0.89	0.22	0.17	4.07 ***	0.64	0.33	0.17	1.96 *
	GEDS	0.11	0.03	0.15	3.69 ***	0.09	005	0.18	2.05 *
	MSPSS	−0.13	0.02	−0.37	−8.78 ***	−0.13	0.03	−0.41	−4.37 ***
	*R*^2^ = 0.22, Adjusted *R*^2^ = 0.21 ***					*R*^2^ = 0.29, Adjusted *R*^2^= 0.26 ***			

Note. ^1^ Male =1, Female = 2; GAD-7 = General Anxiety Disorder; COVID-19 risk = COVID-19 Risk Perception Index; MSPSS = Multidimensional Scale of Perceived Social Support; GEDS = General Ethnic Discrimination Scale. * *p* < 0.05, ** *p* < 0.01, *** *p* < 0.001.

## Data Availability

The datasets used and analyzed during the current study are available from the corresponding author.

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
