# Peer review of "Experience of Ethnic Discrimination, Anxiety, Perceived Risk of COVID-19, and Social Support among Polish and International Students during the Pandemic"

_ijerph, 2023, doi:10.3390/ijerph20075236_

Round 1
Reviewer 1 Report (Previous Reviewer 1)
The author has made several changes, but I still see problems with the research method in line 136, using the survey method or the snowball method. These two terms have different meanings; this must be explained by the author because surveys and snowballs are very different in implementation.
Author Response
Please find the attachment

This manuscript is a resubmission of an earlier submission. The following is a list of the peer review reports and author responses from that submission.
Round 1
Reviewer 1 Report
Snowball technique for quantitative research? Usually, this snowball is used for qualitative research methods. In the limitations section, it is better to explain the limitations of a quantitative study that was not carried out in-depth so that future studies can be conducted with qualitative research to explore the answers to each sample. Why should the selected variables of Ethnic Discrimination, Anxiety, Perceived Risk of COVID-19, and Social Support be explained well? Why about other variables?
Reviewer 2 Report
Thank you for the opportunity to review this study.
The subject addressed is very interesting and up-to-date considering the medical novelty that humanity has gone through in recent years.
It is interesting to note how people were affected psychologically.
The article is well written, the explanations are clear and logical. Also, the presented results are clear, and the discussions are correlated with them.
I am of the opinion that the section intended for limitations should be completed considering the problems that the online questionnaire encounters. There is the possibility of self-selection and, above all, if the redistribution of the questionnaire was allowed (which Google Forms cannot limit), there is the possibility of participants sharing the questionnaire with other people who have the same views, thus certain points of view may be over-represented .
I believe that the article can be published after the section intended for limitations is completed.
